# Frequent and intense human-bat interactions occur in buildings of rural Kenya

**Reilly T. Jackson**[1¤]*, **Tamika J. Lunn**[2,3], **Isabella K. DeAnglis**[1], **Joseph G. Ogola**[4], **Paul W. Webala**[5], **Kristian M. Forbes**[1]

**1** Department of Biological Sciences, University of Arkansas, Fayetteville, Arkansas, United States of America, **2** Odum School of Ecology, University of Georgia, Athens, Georgia, United States of America, **3** Center for the Ecology of Infectious Diseases, University of Georgia, Athens, Georgia, United States of America, **4** Department of Medical Microbiology, University of Nairobi, Nairobi, Kenya, **5** Department of Forestry and Wildlife Management, Maasai Mara University, Narok, Kenya

¤ Current address: Wildlife Research Branch, Arizona Game and Fish Department, Phoenix, Arizona, United States of America

* reillytjackson14@gmail.com

**Data Availability Statement:** All relevant data are within the paper and its Supporting Information files.

## Abstract

Simultaneous use of domestic spaces by humans and wildlife is little understood, despite global ubiquity, and can create an interface for human exposure to wildlife pathogens. Bats are a pervasive synanthropic taxon and are associated with several pathogens that can spill over and cause disease in humans. Urbanization has destroyed much natural bat habitat and, in response, many species increasingly use buildings as roosts. The purpose of this study was to characterize human interactions with bats in shared buildings to assess potential for human exposure to and spillover of bat-borne pathogens. We surveyed 102 people living and working in buildings used as bat roosts in Taita-Taveta county, Kenya between 2021 and 2023. We characterized and quantified the duration, intensity, and frequency of human-bat interactions occurring in this common domestic setting. Survey respondents reported living with bats in buildings year-round, with cohabitation occurring consistently for at least 10 years in 38% of cases. Human contact with bats occurred primarily through direct and indirect routes, including exposure to excrement (90% of respondents), and direct touching of bats (39% of respondents). Indirect contacts most often occurred daily, and direct contacts most often occurred yearly. Domestic animal consumption of bats was also reported (16% of respondents). We demonstrate that shared building use by bats and humans in rural Kenya leads to prolonged, frequent, and sometimes intense interactions between bats and humans, consistent with interfaces that can facilitate exposure to bat pathogens and subsequent spillover. Identifying and understanding the settings and practices that may lead to zoonotic pathogen spillover is of great global importance for developing countermeasures, and this study establishes bat roosts in buildings as such a setting.

## Author summary

Many bat species share space with people and domestic animals. Bat use of buildings, which serve as roosts where they raise young and sleep, is increasing globally with the loss

**Funding:** This work was supported by the Arkansas Biosciences Institute (KMF and TJL) and the University of Arkansas' Sturgis International Fellowship (RTJ). Funders had no role in study design, data collection, decision to publish, or preparation of the manuscript.

**Competing interests:** The authors have declared that no competing interests exist.

of natural habitat. In such settings, interactions between bats, people, and domestic animals may occur. This may be cause for concern as bats can carry pathogens that infect people or domestic animals and interactions with bats may create opportunities for exposure to these pathogens. Until now, the characteristics of human-bat interactions in buildings, and their potential for exposing humans and domestic animals to bat-borne parasites, were unknown. In our study, people living and working in buildings used simultaneously by bats reported frequent interactions with bats and their feces, which can facilitate human exposure to bat pathogens. These interactions happened frequently and over many years. We demonstrate that shared building use by bats and people in rural Kenya leads to prolonged, frequent, and sometimes intense interactions between bats, humans, and domestic animals, consistent with exposure opportunities that can lead to pathogen spillover. Identifying the settings that may lead to human contact with pathogens is critical for developing countermeasures to mitigate public health hazards.

## Introduction

Emerging infectious diseases (EIDs) are a significant threat to global health and security, as demonstrated by the recent COVID-19 pandemic and Mpox disease outbreak [1–3]. Most EIDs have zoonotic origins and emerge in humans via spillover of pathogens from animals, often wildlife [4]. These risks are exacerbated by growing human populations and conversion of natural lands to anthropogenic regions, which can increase human contact with wildlife and exposure to their pathogens [4–6].

Settings and practices that lead to pathogen spillover are little understood but of great importance for informing outbreak mitigation strategies. In lieu of direct knowledge on pathogen exposure, which is extremely difficult to identify from wild animals, characterization of human-wildlife contact can be used to infer exposure risk. Identifying exposure settings has primarily focused on direct contact between humans and wildlife, largely in the form of wildlife hunting and markets for the sale of live animals [7–10]. For example, wildlife consumption and associated handling and butchering creates human contact with wildlife viscera and bodily fluids, which can facilitate spillover of their pathogens [11]. However, contacts between humans and wildlife occur across numerous settings outside of wildlife trade and consumption and can result in human exposure to wildlife pathogens [12]. Other settings and practices that promote contact between wildlife and humans have received far less focus despite the importance of their characterization to mitigating zoonotic pathogen spillover.

Wildlife often share space with humans and domestic animals, especially in the Global South, where humans and wildlife coexist closely in developing landscapes and EID risk is high [13–14]. Studies have reported many communities struggling to manage small mammal incursion into buildings [15–17]. The presence of small mammals in these spaces can create opportunities for human and domestic animal contact with wildlife and their excreta, potentially exposing them to wildlife-borne pathogens [18]. Despite the risk, characterization and quantification of contacts within buildings, where people may spend significant portions of their lives, is lacking.

Bats can harbor zoonotic pathogens that may be shed in excreta and bodily fluids (eg., feces, urine, saliva, blood, etc.) [19–20]. Several bat-borne viruses have emerged in humans after transmission from bats via indirect contact with bat excreta or direct contact with bat bodily fluids [21–24]. Domestic animals can also be exposed to these pathogens after contact with bats and their excreta or fluids [25]. In developing settings, buildings, like family homes, places of worship, and schools, can be highly permeable to bats, and with ongoing habitat loss

bats are increasingly using these structures as roosts [26–27]. Few options exist for people to safely manage bat use of their buildings, and this provides numerous opportunities for human-bat contact and conflict. However, detailed characterization of how humans contact bats and their excreta in relation to pathogen exposure risk in shared spaces is lacking and requires attention.

We investigated human-bat interactions in buildings in rural southwestern Kenya to characterize and quantify forms of contact that could lead to human exposure to bat pathogens. Bats will roost frequently in buildings simultaneously used by humans across Africa [28–31], including the focus area [32], and this region of Kenya is forecasted as a hotspot for zoonotic pathogen emergence where surveillance and mitigation efforts are needed [14]. By understanding interactions between humans and bats and their potential to facilitate pathogen exposure and spillover, we can better identify human health risks and develop evidence-based strategies towards mitigation.

## Methods

### Ethics statement

This research was approved by the Kenyan National Commission for Science, Technology and Innovation (#NACOSTI/P/21/9267), the Kenya Wildlife Service (KWS/BRM/500 and WRTI/RP/118.6), and the University of Arkansas Institutional Review Board (Protocol #2103320918). Participants were informed about the study and verbal consent was obtained prior to conducting surveys.

### Study area

This study was conducted in Taita-Taveta County (Mwatate, Wundanyi, and Voi subcounties), Kenya. The most recent population estimate of Taita-Taveta County was 340,671 people in 2019 [33], with a 1.8% annual increase in population size over the preceding 10 years. Almost three-quarters of the population is considered rural, although urbanization and deforestation are increasing substantially in the region [34–35]. This area is characterized by remnant patches of high-elevation cloud forest surrounded by low-elevation grasslands, woodlands, and agriculture [36].

### Survey methods

We surveyed people in rural and urban regions of Taita-Taveta County during 2021 (August–October), 2022 (January–April), and 2023 (May–June) to understand and characterize human and domestic animal interactions with bats living in buildings. Participants were identified via snowball sampling by engaging in word-of-mouth conversations with community members throughout the study area. As public attitudes towards bats are often negative in Kenya, largely due to the cultural association of bats and witchcraft [29,37], we endeavored to become familiarized and trusted by the community prior to conducting surveys and conducted all surveys with a local field assistant. To initiate the survey process, at least one of the authors and a local Taita assistant entered an area and spoke with members about the presence of bats in nearby buildings. With help from the community, we sought out adults who had bats in their homes (permanent and rental properties) or workplaces at the time of the survey, or who had evidence of recent sustained bat use (i.e., urine staining, fecal deposits, dead bats, etc.). Surveys were directed to one individual per property, however additional family members were sometimes present during questioning.

Surveys were conducted in the local Taita language, Swahili, or English by local Taita assistants and at least one of the authors. Questions were read to respondents by the research team

and answers were transcribed by the team. Our survey consisted of short-answer, dichotomous, and categorical questions to characterize resident human and domestic animal demographics of the property, the duration of bat use of the property and its buildings, and human and domestic animal interactions with bats and their excreta on the property (see Supplementary Materials for detailed information on survey questions). Surveys from 2021 ($n = 23$) included 23 multi-part questions. After this initial data collection, we added one additional question to characterize human and domestic animal contact with dead bats on the property. Therefore, surveys conducted in 2022 and 2023 ($n = 79$) included 24 multi-part questions.

## Data Analysis

To explore the effect of the number of residents on the property, length of bat building use, and respondent demographics (gender, education, and age) on direct (e.g., touching, scratches, bites, etc.) and indirect (e.g., contact with bat excrement) interactions with bats, we used univariate generalized linear models with a binomial error distribution and logit link function. Surveys with incomplete data were excluded for individual demographic measurements. We used chi-square tests to compare the frequencies of bat interactions, duration of bat occupation of buildings, bat exclusion methods employed by inhabitants, and reasons for exclusion. All analyses were conducted in R (Version 2023.06.2+561) using the stats package (v4.1.3).

## Results

We surveyed 102 indigenous Taita people who lived or worked in buildings used by bats (S1 Table). Over 70% of people reported bat use of their buildings for >5 years ($n = 72$), with bat presence for 5–10 years most commonly reported ($\chi^2 = 36.52$, $P < 0.01$, Fig 1). Most properties (88%) had bat presence year-round ($n = 90$). Survey participants described frequent exposure to bats that would support pathogen transmission through two main routes: direct and indirect (fecal/oral) contact, with indirect contact between bats and humans reported more frequently than direct contacts ($\chi^2 = 24.77$, $P < 0.01$, Fig 2A).

Close to half of participants (39%) reported direct contact with bats, including people touching bats ($n = 40$) and one report of being bitten. People on the property (children,

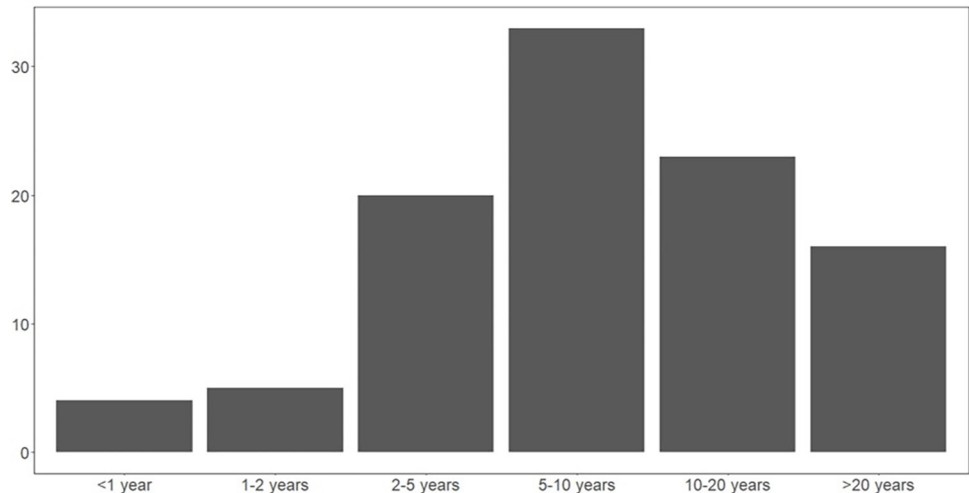

**Fig 1. The number of respondents reporting the length of time bats had been present in their buildings.** All but one participant answered this question.

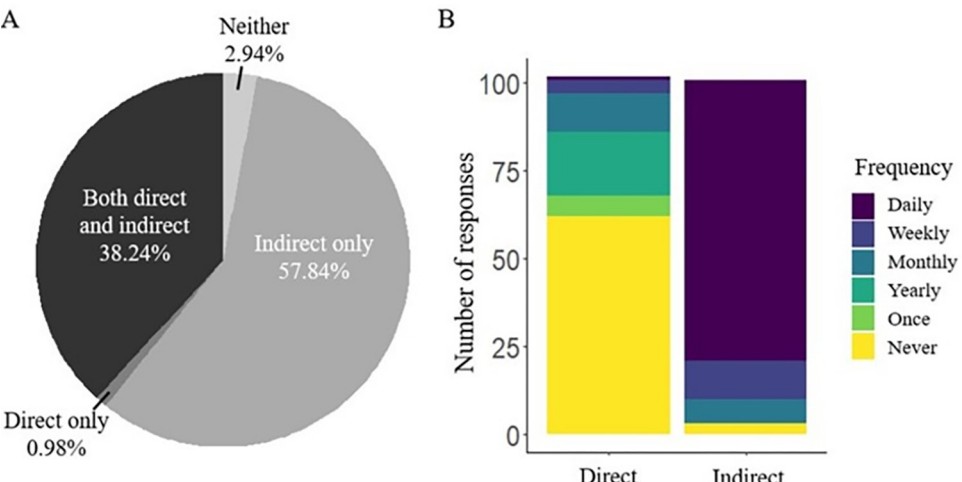

**Fig 2. People surveyed about human-bat interactions in their buildings reported both direct (e.g., touching, scratches, bites) and indirect (e.g., contact with bat excrement) contacts between humans and bats.** (A) Indirect interactions were the most reported of the two interaction types; (B) Frequency of these contacts varied from daily interactions to never having these interactions.

spouses, custodians) other than the respondent engaged in these interactions as well. Direct interactions occurred with varying frequencies over time according to respondents (Fig 2B). Reports of direct contact did not differ significantly based on the number of residents on the property, length of time of bat use, or respondent demographics (gender, education, and age; $P > 0.06$).

Over 90% of respondents reported indirect contact with bats, mostly through interactions with their feces and urine ($n = 98$). Daily occurrences of indirect contact were reported by most participants (78%, $\chi^2 = 285.06$, $P < 0.01$, Fig 2B) and children, spouses, house guests, and custodians were also involved in these interactions. Reports of indirect contact did not differ significantly based on the number of residents on the property, length of time of bat use, or respondent demographics (gender, education, and age; $P > 0.13$).

Attempts to remove bats from buildings were reported by almost 80% of participants ($n = 81$). Of those reporting removal efforts, almost half reported direct contact with bats ($n = 34$). Numerous removal methods were reported, with fumigation via pesticide, blocking access to building entry points, and scaring bats from buildings reported more than other methods ($\chi^2 = 107.37$, $P < 0.01$; Table 1). Bats returned to the building after removal efforts in

**Table 1. Methods used by people to remove bats from buildings.** Attempts to remove bats were common and frequently led to direct contact with bats that could facilitate pathogen exposure. This question allowed for multiple responses from respondents.

| Type of removal effort | Number of responses (%) |
| --- | --- |
| Fumigation via pesticide | 36 (44.44) |
| Blocking access to buildings | 25 (30.86) |
| Scaring bats | 22 (27.16) |
| Killing individual bats | 19 (23.46) |
| Smoking bats out | 5 (6.17) |
| Application of holy water | 2 (2.47) |
| Removal of ceiling | 2 (2.47) |
| Application of salt | 2 (2.47) |
| Killing via domestic animal | 1 (1.23) |

over 90% of cases ($n = 76$). Bad smells ($n = 39$), noise ($n = 39$), dirt from feces and urine ($n = 36$), and damage to property ($n = 21$) were the most common reasons reported for removing bats. Compared to more common removal reasons, significantly fewer respondents mentioned worries about witchcraft ($n = 15$), that bats were a general nuisance ($n = 12$) or posed health risks to people ($n = 8$; $\chi^2 = 123.47$, $P < 0.01$).

We asked a subset of participants about the presence of dead bats on their properties ($n = 79$). Nearly 65% reported dead bats on properties ($n = 51$). Most removed dead bats, usually by throwing them over property lines ($n = 30$) or swept them outside ($n = 7$). Some also reported burning ($n = 6$), feeding domestic cats ($n = 2$), and burying bat carcasses ($n = 1$). Interestingly, 13 respondents reported seeing domestic animals (dogs, cats, and chickens) consume dead bats on their property, most often their own animals.

## Discussion

This is the first study to focus on buildings as an important interface for human-bat interactions and demonstrates that there are pathways for human exposure to bat-borne pathogens in these settings. We establish that human-bat contacts in rural East Africa are common in buildings and that these interactions can be intense, frequent, and occur consistently over long periods of time. Our survey respondents had exposure to bats in ways that can promote pathogen transmission through direct or indirect pathways, as well as via domestic animals as intermediate hosts. Much attention has focused on bushmeat hunting and wet markets as high-risk practices and settings for wildlife pathogen exposure risk. Given the increasing rate of urbanization and subsequent habitat loss bats are experiencing, the sharing of anthropogenic structures by humans and bats is likely to become more common across the globe, thereby increasing the risk of zoonotic spillover.

Our results show that bats and humans contacted each other directly (e.g., touching, scratches, bites, etc.) and indirectly (e.g., contact with bat excrement), with both pathways presenting concerns for public health. Direct contacts can expose humans to lethal viruses hosted by bats, with various lyssaviruses (including rabies virus) being the most well-known bat-borne pathogens transmitted in this manner [38]. Indirect contacts were frequently reported in our study and are also common pathways for zoonotic pathogen spillover [39]. Bat excreta reported in these indirect interactions, mostly feces, can contain pathogens shed by bats, including coronaviruses, rotaviruses, and paramyxoviruses that are viral families of human health concern [20]. Fungal pathogens, like *Histoplasma capsulatum*, the causative agent of histoplasmosis, may also be inhaled from bat fecal dust and have infected people living in buildings with bat roosts in Africa [40].

Multiple respondents reported observing domestic animals–mainly cats, dogs, and chickens–consuming bats. Predation and consumption of bats can facilitate transmission of zoonotic pathogens into consumers, including domestic animals [41], which can also serve as bridge or intermediate hosts for onward transmission to humans [42]. Furthermore, bats often roost in livestock enclosures in this region and may deposit feces or bodily fluids in spaces frequently used by domestic animals [43]. Many frugivorous bat species chew and eject saliva-covered fruit pulp below their roosts, which domestic animals may then consume and become exposed to shed pathogens [44]. Indeed, it is thought that Nipah and Hendra virus, both paramyxoviruses with high human mortality, emerged in pigs and horses in this fashion, respectively [25].

As described under the OneHealth framework, the health of domestic animals and humans depends on ecosystem quality, which is a function of wildlife health [45]. In addition to human disease risks associated with bats, our results show that bat individuals and populations

may be negatively impacted by their interactions with humans in buildings. Respondents attempted to remove bats from their buildings, mostly via fumigation with pesticides, blocking bat entrance points, and direct killing of bats. Paradoxically, these activities often led to direct human contact with bats, creating additional opportunities for pathogen exposure. Stress to bats caused by removal attempts can also increase pathogen transmission risk by altering bat behavior and immune function, which collectively drive contact rates and pathogen susceptibility and shedding [46–48]. Furthermore, high bat mortality can negatively impact the critical ecosystem services that bats provide by reducing their ability to consume insect pests, pollinate fruit trees, and disperse seeds [49].

It is worth noting that this study had a relatively small sample size and employed a survey, which required respondents to opt in for participation. In Kenya, there is negative cultural stigma associated with people interacting with bats, as bats are often maligned as witches, bad omens, or general harbingers of malaise [29]. Occasionally, humans with bats in their buildings declined to participate in the survey, potentially due to these cultural stigmatizations. It is possible that our respondents were biased towards people with strong opinions or greater interactions with bats that were willing to share more of their experiences, or that did not care or were unaffected by regional customs about bats [50]. While sample size and survey methods may limit the inference of our results, the identification and characterization of documented interactions demonstrate that relevant pathways for zoonotic pathogen exposure and spillover are common in these settings.

Across Africa, bats are frequently found in anthropogenic structures where there is increased likelihood of human exposure to bats and, consequently, their pathogens [28–31]. Measures mitigating human-bat contact in such settings, such as structural modifications to existing structures that reduce the likelihood of bat use, the construction of buildings inaccessible to bats, or campaigns educating the public about the importance of (and public health concerns associated with) bats, may be instrumental in reducing risk for human exposure to bat-borne pathogens [29,43,51]. Previous work in Taita-Taveta county suggests that modification of building microclimate and proper sealing of buildings, especially in modern tall, cement-walled structures, may reduce bat presence [32,43]. Continent-wide efforts to similarly adjust housing attributes based on local bat selection parameters may be beneficial to reducing bat occupancy in buildings. Furthermore, increasing educational discussions, such as region-specific conversations addressing community mitigation needs, the nature of human-bat relationships, and methods for safe interactions, is also key to curtailing human contact with bats in buildings, ultimately reducing the risk of pathogen exposure and human-induced bat mortality.

The presence of bats in buildings is common in developing regions and our findings establish that there are frequent and prolonged interactions between humans and bats in these settings, consistent with interactions that can facilitate pathogen spillover. Bat mortality is also frequent in these settings, with further ramifications for increased bat-human contacts and decreased quality of wildlife and ecosystem health. Therefore, future community-driven research within a OneHealth framework that explores the impacts of co-habitation on humans, domestic animals, and bats, will be important to assessing the general health risks of these environments. Given the increasing rate of urbanization and subsequent habitat loss bats are experiencing, anthropogenic structure sharing by humans and bats is likely to become more common globally and a greater risk setting for zoonotic pathogen spillover. [26–27].

## Supporting information

**S1 Table. Demographics of respondents asked about their interactions with bats living in their buildings in rural Kenya.** Data from these demographics were incorporated into

analyses to understand risk factors for direct and indirect interactions between humans and bats in anthropogenic structures.
(DOCX)

**S1 Materials. This file contains 1) the English questionnaire used during surveys of people living and working with bats in their buildings, and 2) the demographic data of people surveyed for this study.**
(DOCX)

**S2 Materials. This file contains the data used in this study.**
(XLSX)

**S3 Materials. The files contains the completed STROBE checklist, showing this study's adherence to observational study guidelines.**
(DOCX)

## Acknowledgments

We thank Peter Mwasi, Benson Lombo, and Darius Kimuzi for their assistance in data collection. We thank and acknowledge the Taita people, the stewards of the land where this study was conducted, for their enthusiasm in working with us and describing their experiences. We also thank the Taita Environmental Research and Resource Arc for their logistical assistance, especially Miltone Kimori and Ken Gicheru. We thank Dr. David Irungu for his assistance with translation of the survey.

## Author Contributions

**Conceptualization:** Reilly T. Jackson, Joseph G. Ogola, Paul W. Webala, Kristian M. Forbes.

**Data curation:** Reilly T. Jackson, Tamika J. Lunn, Isabella K. DeAnglis, Joseph G. Ogola, Paul W. Webala, Kristian M. Forbes.

**Formal analysis:** Reilly T. Jackson.

**Funding acquisition:** Reilly T. Jackson, Tamika J. Lunn, Kristian M. Forbes.

**Investigation:** Reilly T. Jackson, Isabella K. DeAnglis, Joseph G. Ogola, Paul W. Webala, Kristian M. Forbes.

**Methodology:** Reilly T. Jackson, Joseph G. Ogola, Paul W. Webala, Kristian M. Forbes.

**Project administration:** Reilly T. Jackson, Tamika J. Lunn, Kristian M. Forbes.

**Supervision:** Kristian M. Forbes.

**Writing – original draft:** Reilly T. Jackson.

**Writing – review & editing:** Reilly T. Jackson, Tamika J. Lunn, Isabella K. DeAnglis, Joseph G. Ogola, Paul W. Webala, Kristian M. Forbes.

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
