## [Decision Letter · Decision Letter 0]

2 Jan 2024

Dear Ms Jackson,

Thank you very much for submitting your manuscript "Buildings promote frequent and intense contact between humans and bats in rural Kenya" for consideration at PLOS Neglected Tropical Diseases. As with all papers reviewed by the journal, your manuscript was reviewed by members of the editorial board and by several independent reviewers. The reviewers appreciated the attention to an important topic. Based on the reviews, we are likely to accept this manuscript for publication, providing that you modify the manuscript according to the review recommendations. 

Three reviewers have evaluated your manuscript and all three indicated that it is well written and presents valuable information and understanding about human-bat interactions. All three reviewers did ask for minor revisions to clarify certain aspects of the manuscript and each provided several suggestions for improvement. Please evaluate these comments, revised the manuscript accordingly and resubmit. Thank you!

Sincerely,

Richard A. Bowen

Academic Editor

Justin Remais

Section Editor

Three reviewers have evaluated your manuscript and all three indicated that it is well written and presents valuable information and understanding about human-bat interactions. All three reviewers did ask for minor revisions to clarify certain aspects of the manuscript and each provided several suggestions for improvement. Please evaluate these comments, revised the manuscript accordingly and resubmit. Thank you!

Reviewer's Responses to Questions

**Key Review Criteria Required for Acceptance?**

**Methods**

-Are the objectives of the study clearly articulated with a clear testable hypothesis stated?

-Is the study design appropriate to address the stated objectives?

-Is the population clearly described and appropriate for the hypothesis being tested?

-Is the sample size sufficient to ensure adequate power to address the hypothesis being tested?

-Were correct statistical analysis used to support conclusions?

-Are there concerns about ethical or regulatory requirements being met?

Reviewer #1: The study sample size of 102 people is small to test the hypothesis given. The authors make reference to the total human population of Taita Taveta county. However the nature of the county landscape that is under protected forests means that human population are congregated around certain areas. The authors also need to give a better description of the infested buildings for example what were the roofs made of (iron sheets or brick tiles) did the building material play a role in raising the risk for bat inhabiting buildings? 

The term rural Kenya is misleading as the county governance structure is devolved meaning the counties have clear urban and rural areas. Was the study conducted in an Urban area like Taita Taveta town? This means that the population also sampled would not be indigenous to the county but people from other parts of Kenya who settle in the town as they are employed or run enterprises and thus require the build rental houses. The study authors need to disaggregate study respondents in two groups indigenous to the county and not indigenous. Literacy level also plays a role in how people approach and tackle issues why was this not a variable on why the respondents chose to live in a build house instead of the traditional roof thatched houses (that may or may have not been also inhabited by bats-did the study come across thatched houses inhabited by bat colonies?)

Reviewer #2: Adequate methods.

Reviewer #3: Objectives were clear enough but the method of identifying the population of individual residences with bat roost/bats living on the property was certainly not clear. It sounds like snowball sampling was the general method, as they describe as 'word of mouth' but the recruitment method needs to be elaborated upon. 

Sample size is insufficient, or more description is needed for explaining those who were approached and refused.

**Results**

-Does the analysis presented match the analysis plan?

-Are the results clearly and completely presented?

-Are the figures (Tables, Images) of sufficient quality for clarity?

Reviewer #1: The results are clear and match the analysis plan put forward in the methods section of the paper. However, in Table 1. (Methods used by people to remove bats from buildings). The responses total is 114 yet there were 102 respondents can the authors explain if this was part of a multiple response data analysis?

Reviewer #2: Adequately presented results.

Reviewer #3: Results for this building study are okay, but could be broadened.

**Conclusions**

-Are the conclusions supported by the data presented?

-Are the limitations of analysis clearly described?

-Do the authors discuss how these data can be helpful to advance our understanding of the topic under study?

-Is public health relevance addressed?

Reviewer #1: The study describes well the zoonotic risk exposure pathways. This is a commendable effort as this was a known fact that had not been documented or quantified. However, the authors missed the opportunity to give a clear policy direction on what type of building design and materials would deter bat infestation. In addition, the failure of authors to describe accurately the unique ecosystem of Taita Taveta means there was a missed opportunity to direct future research to the county as it may be one of the few natural habitats for bats in Kenya.

Reviewer #2: -Are the conclusions supported by the data presented? 

Yes

-Are the limitations of analysis clearly described?

Yes, the study discuss the limitations of being survey-based.

-Do the authors discuss how these data can be helpful to advance our understanding of the topic under study?

I think the discussion could develop more on this.

-Is public health relevance addressed?

Yes, but I think this study could discuss further on what can be done in the local context to reduce or mitigate bat-human conflicts, especially after mentioning increase in risk in the foreseeable future.

Reviewer #3: There is no Conclusion section, just Discussion that ends the paper. Conclusions could be expanded to talk about potential risk mitigation measures on the number of issues raised in Results section.

Authors do not discuss how data can be helpful to advance bat exposure to humans.

and Public health relevance is NOT addressed.

**Editorial and Data Presentation Modifications?**

Reviewer #1: Minor edit

Title: Include the county name Taita Taveta and if possible the exact geographical area the study was conducted (sub county and village). The title misrepresents the highly diverse nature of remote locations in Kenya. Taita Taveta has the highest number of protected forests,48, and does not represent the ecology of the arid and semi-arid areas which make up 80% of Kenya's land mass. 

Could this be also the reason there are high bat infestation because of the high forest cover? 

In the abstract section consider revising the third sentence from the top this will improve clarity. 

Bats are associated with several pathogens that can spill over and cause diseases in humans. Rapid urbanization has resulted in the encroachment and loss of the bats' natural habitat, forcing the bats to use anthropogenic structures for roosting.

Reviewer #2: L188 - I recommend inverting the sentence, start with `Compared to more common removal reasons,`

Figure 1 - Maybe report somehow in this figure the age of respondents for each category or the amount of time respondents have been living at the same place and discuss how those numbers could approximate to a good estimate.

Reviewer #3: THis paper has merit but needs further work. there are topics like the domestic animals' eating of dead bats that should be expanded upon, and this inclusion in 'building'/ediface paper makes me think there could be a follow-up suggested to expand on this narrowly focused questionnaire data than the team is exploiting. Would be a richer paper with a more full-bodied approach and exploration of some of these interface dynamics. Since I see the questionnaire does not include other interface questions, I would heartily suggest further development of results and Discussion section, perhaps suggesting next steps on avenues for exploration, and adding a Conclusions section.

**Summary and General Comments**

Reviewer #1: The study has notable strengths in documenting and quantifying possible risk pathways for humans sharing living spaces with bats. However the small sample size makes this more an observational study that does not give a high statistical power to make inferences from. 

In the discussion section, the authors highlight a study bias of selecting respondents who were unaffected by regional customs about bats. The study failed to describe the study population (indigenous or migrants) so this statement needs to be removed. In addition, if the survey was conducted in an urban areas the most likely respondents were not indigenous Taita Taveta community who would be inclined to the cultural fear of talking about bats . The respondents would have been workers from other parts of Kenya who will not be affected by the local cultural belief system. 

In conclusion, if the authors have more data on the type of housing associated with bat infestations. They can analyse this data and make a policy recommendation on the design and type of roofing that can deter bat roosting especially in an area that is a hot spot for closer bat-human coexistence due the nature of the unique conserved forest landscape of Taita Taveta County.

Reviewer #2: Dear authors, 

Studies like this are much needed so public health authorities can promote guidelines for mitigating bat-human conflict. I believe this draft requires minor changes in order to be considered for publication in PlosNTD. 

The suggested changes would be focused on the discussion. For instance, the mentioned stigma could be target of educational projects that could be applied in the region, and some discussion on this topic would be desired. Moreover, I feel like the authors could illustrate more the local context in terms of mentioning existing community projects acting in the region, if there were attempts to provide guidelines for living peacefully with bats. Moreover, the discussion could explore future research demands in the region. Finally, I felt this study could mention how to integrate surveys like this to the One Health approach, by briefly discussing links between human health and biological conservation.

Reviewer #3: This paper seems cursory, and a bit superficial, but is of scientific interest, as not much has been done on the bat/human co-habitation interface.

PLOS authors have the option to publish the peer review history of their article (what does this mean?). If published, this will include your full peer review and any attached files.

Reviewer #1: No

Reviewer #2: No

Reviewer #3: No

Figure Files:

Data Requirements:

Reproducibility:

References

---

## [Editor Report · Decision Letter 1]

12 Feb 2024

Dear Dr Jackson,

We are pleased to inform you that your manuscript 'Frequent and intense human-bat interactions occur in buildings of rural Kenya' has been provisionally accepted for publication in PLOS Neglected Tropical Diseases.

Best regards,

Richard A. Bowen

Academic Editor

Justin Remais

Section Editor

---

## [Editor Report · Acceptance letter]

22 Feb 2024

Dear Dr Jackson,

We are delighted to inform you that your manuscript, "Frequent and intense human-bat interactions occur in buildings of rural Kenya," has been formally accepted for publication in PLOS Neglected Tropical Diseases.

Best regards,

Shaden Kamhawi

co-Editor-in-Chief

Paul Brindley

co-Editor-in-Chief
